# Transport of Non-Steroidal Anti-Inflammatory Drugs across an Oral Mucosa Epithelium In Vitro Model

**DOI:** 10.3390/pharmaceutics16040543

**Published:** 2024-04-15

**Authors:** Grace C. Lin, Heinz-Peter Friedl, Sarah Grabner, Anna Gerhartl, Winfried Neuhaus

**Affiliations:** 1Competence Unit Molecular Diagnostics, AIT Austrian Institute of Technology GmbH, 1210 Vienna, Austriaanna.gerhartl@gmail.com (A.G.); 2Department of Medicine, Faculty of Medicine and Dentistry, Danube Private University, 3500 Krems, Austria; 3Division of Pharmaceutical Technology and Biopharmaceutics, University of Vienna, 1090 Vienna, Austria

**Keywords:** inflammation, TR146, blood–saliva barrier, non-steroidal antiphlogistic drugs

## Abstract

Non-steroidal anti-inflammatory drugs (NSAIDs) are one of the most prescribed drugs to treat pain or fever. However, oral administration of NSAIDs is frequently associated with adverse effects due to their inhibitory effect on the constitutively expressed cyclooxygenase enzyme 1 (COX-1) in, for instance, the gastrointestinal tract. A systemic delivery, such as a buccal delivery, of NSAIDs would be beneficial and additionally has the advantage of a non-invasive administration route, especially favourable for children or the elderly. To investigate the transport of NSAIDs across the buccal mucosa and determine their potential for buccal therapeutic usage, celecoxib, diclofenac, ibuprofen and piroxicam were tested using an established oral mucosa Transwell^®^ model based on human cell line TR146. Carboxyfluorescein and diazepam were applied as internal paracellular and transcellular marker molecule, respectively. Calculated permeability coefficients revealed a transport ranking of ibuprofen > piroxicam > diclofenac > celecoxib. Transporter protein inhibitor verapamil increased the permeability for ibuprofen, piroxicam and celecoxib, whereas probenecid increased the permeability for all tested NSAIDs. Furthermore, influence of local inflammation of the buccal mucosa on the transport of NSAIDs was mimicked by treating cells with a cytokine mixture of TNF-α, IL-1ß and IFN-γ followed by transport studies with ibuprofen (+ probenecid). Cellular response to pro-inflammatory stimuli was confirmed by upregulation of cytokine targets at the mRNA level, increased secreted cytokine levels and a significant decrease in the paracellular barrier. Permeability of ibuprofen was increased across cell layers treated with cytokines, while addition of probenecid increased permeability of ibuprofen in controls, but not across cell layers treated with cytokines. In summary, the suitability of the in vitro oral mucosa model to measure NSAID transport rankings was demonstrated, and the involvement of transporter proteins was confirmed; an inflammation model was established, and increased NSAID transport upon inflammation was measured.

## 1. Introduction

The discovery of salicylic acid in the 19th century led to the formulation of one of the most popular drugs to treat inflammation and pain, acetylsalicylic acid, also known as aspirin [1]. Since then, several substances of various chemical structures with anti-inflammatory and antipyretic properties have been discovered and are classed as non-steroidal anti-inflammatory drugs (NSAIDs) [2]. NSAIDs have the advantage of lesser adverse side effects compared to other analgesics, such as opioids [3,4,5], and remain one of the most described drugs worldwide [6]. Their therapeutic effect is due to the inhibition on cyclooxygenase 1 and 2 (COX), of which COX-1 is constitutively expressed in the tissue (i.e., gastrointestinal tract) and described as a housekeeping gene, whereas COX-2 is upregulated upon inflammation [7]. The COX enzymes catalyse the conversion of arachidonic acid into thromboxane and prostaglandin, eicosanoids with pro-inflammatory activities on the site of inflammation [8,9,10,11]. However, the functionality of prostaglandins are very versatile, as they also play a protective role in the gastrointestinal tract [10]. Hence, inhibition of COX-1 compromises the integrity of the gastrointestinal mucosa, leading to adverse effects such as gastrointestinal bleeding or stomach ulcers [12]. For this reason, NSAIDs are grouped into non-selective NSAIDs (ibuprofen, diclofenac, piroxicam) and COX-2 selective NSAID (celecoxib). Additionally, due to their diverse chemical structures, they are also grouped accordingly into salicylic acid derivates (i.e., aspirin), propionic acid derivates (i.e., ibuprofen, naproxen), acetic acid derivatives (i.e., indomethacin), enolic acids (i.e., piroxicam) or diaryl heterocyclic acids (i.e., celecoxib), among others [9,13,14]. Up to date the majority of NSAIDs on the market belong to the group of non-selective NSAIDs [9]. Additionally, COX-3, a splice variant of COX-1, expressed in the heart and cerebral cortex, has shown to be inhibited by paracetamol (more commonly known as acetaminophen) [15]. However, findings of COX-3 inhibition in humans are inconclusive [16].

With regard to the adverse effect on the gastrointestinal mucosa, circumventing the gastrointestinal tract and first-pass metabolism via systemic delivery of NSAIDs is beneficial, especially for patients requiring long-term treatment. While drugs are frequently administered via the intravenous route for this purpose, injections are often associated with discomfort and require trained personnel. Especially children are affected by fear of needles; however, adults have also been reported to delay medical treatments due to traumatic experiences with needles [17,18]. For these reasons, delivery by buccal absorption is preferable next to sublingual absorption, as the highly vascularised oral mucosa offers the possibility of non-invasive drug delivery while circumventing the gastrointestinal tract. In recent years, the suitability of biomaterials for regulated drug release has been studied intensively. Until now, several studies describing desirable composition and formulation of biomaterials (i.e., based on hydrogels or cellulose) for drug delivery have been published and summarised [19,20]. Recent studies with an NSAID as therapeutic agent describe buccal delivery systems for ibuprofen and indomethacin based on lipid nanoparticles loaded on mucoadhesive hydrogels and cellulose film [21,22]. To date, several buccal formulations with a wide range of drugs to treat insomnia or angina have been introduced to the market, including products based on opioids fentanyl and sufentanil for analgesic treatment [19]. However, marketed buccal formulations for pain treatment based on NSAIDs remain limited. Even though the buccal delivery route offers the advantage of bypassing the gastrointestinal tract, the effective absorption of drugs with low lipid solubility or larger molecular size is often limited and most drugs are assumed to cross the barrier using the paracellular or transcellular route by passive diffusion [23].

In this study, we aimed to investigate the permeability of NSAIDs of various chemical structures (acetic acid derivatives: diclofenac, propionic acid derivatives: ibuprofen, enolic acid derivatives: piroxicam) and selectivity (COX-2 selective: celecoxib) in more detail to evaluate their potential for buccal delivery. For this purpose, we used a thoroughly characterised model of the human buccal mucosa, which has been previously demonstrated to be suitable to study transport mechanisms of larger molecules [24,25]. Furthermore, we aimed to evaluate the influence of multidrug resistance transporter inhibitors probenecid and verapamil on the permeability of NSAIDs, as they have shown to increase the anti-inflammatory effect or transport of dauricine or NSAIDs in previous studies [26,27,28]. Additionally, we aimed to induce an inflammatory response by exposure to pro-inflammatory cytokines and assess the influence of the inflammation on the permeability of the selected NSAID ibuprofen.

## 2. Materials and Methods

### 2.1. Cell Culture

Human buccal carcinoma cells TR146 were purchased from Sigma-Aldrich (St. Louis, MO, USA), (10032305) and cultivated as described in detail previously [24]. In short, cells were cultivated in Dulbecco’s Modified Eagle Medium (DMEM, Sigma-Aldrich, D5796) supplemented with 10% Fetal Calf Serum (Sigma-Aldrich, F9665) and 1% Penicillin/Streptomycin (Merck, St. Louis, MO, USA, A2213) at 37 °C and propagated weekly at a cell density of 9.33 × 10^3^/cm^2^. For transport studies, cells were seeded at a density of 4.29 × 10^4^/cm^2^ in cultivation media on 24-well ThinCerts (Greiner, Kremsmünster, Austria, 662641), and for Western blot analysis on 6-well ThinCerts (Greiner, 657641). As soon as cells reached confluency, cultivation was switched from submerged to airlift and 1% Human Keratinocytes Growth Supplements (HKGS, Gibco, Thermofisher, Vienna, Austria, S0015) was added to the cultivation media on the basolateral compartment. Media were changed every 2–3 days until the day of experiment on day 29. Cells from passage 14–36 were used for experiments. Transepithelial electric resistance (TEER) was measured on day of experiment, as described in detail previously [25,29].

### 2.2. Transport Study

Piroxicam (P5654), ibuprofen (I1892) and probenecid (P8761) were purchased from Sigma-Aldrich, verapamil (94837) and carboxyfluorescein (21877) from Fluka, celecoxib, diclofenac and diazepam were a kind gift from Dr. Maierhofer (AGES, PharmMed, Vienna, Austria). Substances were dissolved in DMSO (Roth, Graz, Austria, A994.2), dd-H_2_O, or cultivation media for stock solutions (piroxicam: 100 mM in DMSO; carboxyfluorescein: 2.657 mM in cultivation media; celecoxib: 100 mM in DMSO; diclofenac: 3 mM in dd-H_2_O; ibuprofen: 10 mM in dd-H_2_O; diazepam: 100 mM in DMSO; probenecid: 100 mM in DMSO; verapamil; 100 mM in DMSO), sterile filtered and stored at 4 °C until usage. Solutions were diluted to 10 µM (carboxyfluorescein) or 100 µM in DMEM on day of experiments.

First, TEER was measured in cultivation media on the day of experiment. Afterwards, cells and blanks were washed twice with DMEM on the apical (300 µL) and basolateral (900 µL) compartments and selected cells and blanks were treated with DMEM with or without 100 µM probenecid or verapamil on the apical and basolateral side at 37 °C for 30 min. Prior to the start of the experiment, apical media were replaced with DMEM containing 100 µM diazepam, 10 µM carboxyfluorescein and 100 µM of NSAID (celecoxib, diclofenac, ibuprofen or piroxicam). To cells which were pre-incubated with probenecid or verapamil, 100 µM probenecid or verapamil was applied apically as well. DMSO content was adapted for cells with no inhibitors, and DMSO and dd-H_2_O content was adjusted in the basolateral compartment as well in order to apply the same vehicle concentrations on the apical and basolateral compartment. Cells were incubated in the dark at 37 °C, 5% CO_2_ and 95% humidity, and every 60 min, cells and blank inserts were transferred to fresh basolateral media of the same composition. After 240 min, media from apical and basolateral compartments were collected for analysis and stored at 4 °C together with prepared stock solutions and media blanks. Samples were measured as duplicates (apical samples) or triplicates (basolateral samples, stock solutions, blank solutions) at 488/520 nm excitation/emission wavelength to estimate relative fluorescence units (RFU) of carboxyfluorescein using the Enspire Multimode Plate Reader (PerkinElmer, Traiskirchen, Austria). Subsequently, duplicates of apical sample were collected, and samples were frozen at −80 °C until further analysis with High-Performance Liquid Chromatography (HPLC).

### 2.3. High-Performance Liquid Chromatography

To analyse NSAIDs and diazepam, samples were thawed and precipitated with gradient-grade (≥99.9%) acetonitrile (CAN; VWR, Radnor, PA, USA, 83639.400, HiPerSolv Chromanorm^®^) in a one-part-plus-one-part ratio for at least 1 h at 4 °. After centrifugation for 10 min at 11,384× *g* rpm at 4 °C, supernatants were transferred to 1.5 mL transparent glass vials with 11 mm nominal diameter (Machery-Nagel, Vienna, Austria, 70201HP) and vials were sealed with 11 mm aluminium crimp closures (Machery-Nagel, 702001) using a manual crimper (Thermo Scientific, Vienna, Austria, 4012-100). ACN and 10 mM KH_2_PO_4_ (Sigma-Aldrich, P0662-500G)—adjusted to pH 3 with 10 mM 85% H_3_PO_4_ (Sigma-Aldrich, 49685-100 mL)—were degassified in an ultrasonic bath for at least 30 min prior to usage. Quantification was performed using an UltiMate^TM^ 3000-System (Thermo Fisher Scientific) equipped with HPG-3400SD standard binary pump (Thermo Fisher Scientific, 5040.0041), Dionex^TM^ TCC-3200 column compartment (Thermo Fisher Scientific, 5730.0010), Dionex^TM^ DAD-3000 variable wavelength detector (Thermo Fisher Scientific, 5082.0010) and WPS-3000TSL auto sampler (Thermo Fisher Scientific, 5822.0020). Sample injection volume was 10 µL and separated at a flow rate of 1 mL/min using a BDS Hypersil^TM^ C18 column (Thermo-Fisher, 28105-254630, 250 × 4.6 mm, 5 µm particle size) equipped with a BDS Hypersil^TM^ C18 5 µm 10 × 4 drop-in guard column (Thermo-Fisher, Vienna, Austria, 28105-014001).

Quantification of NSAID solutions with diazepam was performed with isocratic elution with different ratios of acetonitrile (ACN) and buffer (10 mM KH_2_PO_4_/10 mM H_3_PO_4_, pH = 3) except for celecoxib (diclofenac: 47:53 ACN/buffer; piroxicam 50:50 ACN/buffer, ibuprofen 60:40 ACN/buffer). To analyse solutions containing celecoxib and diazepam, diazepam was first detected at 11 min retention time after running with 45:55 ACN/buffer, then the ratio was increased to 80:20 ACN/buffer in a gradient from 12 to 14 min, then continued to run isocratically from 14 to 18 min with 80:20 ACN/buffer; celecoxib was detected at 18 min and ACN was reduced to 45% between 18 and 20 min run time. To reach equilibrium, the column was washed with 45:55 ACN/buffer from 20 to 22 min before the start of the next analysis run. Celecoxib, diclofenac and ibuprofen were detected at 220 nm, diazepam at 254 nm and piroxicam at 350 nm. NSAIDs with diazepam and verapamil were separated in the same manner as NSAIDs and diazepam. Solutions containing NSAIDs, diazepam and probenecid were separated with 35:65 ACN/buffer for 25 min, a gradient from 25 to 30 min to reach 80:20 ACN/buffer and running at 80:20 ACN/buffer from 30 to 34 min. Subsequently, a gradient was applied from 34 to 36 min to reach 35:65 ACN/buffer and the column was further equilibrated at 35:65 ACN/buffer from 36 to 38 min. Samples collected after transport studies, applied stock solutions and background media were measured in triplicates. Peaks were quantified using the software Chromeleon Version 7.2.6, and corresponding mili absorbance unit × time [mAU × min] values were used for data analysis.

### 2.4. Calculation of Permeability Coefficient

Mean values of relative fluorescence unit (RFU) or detected area [mAU × min] were used for calculation. For the first timepoint, the clearance [µL] for the detected area was calculated as shown in the formula below (1a), with A_basot1_ as the measured area [mAU × min] at the first time point of the basolateral sample, V_baso_ as the basolateral volume [µL] and A_stock_ as the area measured for the applied stock solution. For further time points, clearance was similarly calculated as in Formula 1b, subtracting summarised clearance of the previous time point multiplied by the volume factor for basolateral/apical of 3. Clearance [µL] for carboxyfluorescein was calculated with the same formulas using RFU.
(1a)$$Clearance=A_{{baso}_{t_{1}}}\times \frac{V_{baso}}{A_{Stock}}\;$$
(1b)$$Clearance_{t_{n}}=A_{{baso}_{t_{n}}}\times \frac{V_{baso}}{A_{Stock}-\sum \left(A_{baso\left(t_{n-1}\right)\;}\times 3-\;A_{baso_{t_{1}}\;}\times 3\right)}$$

Cleared volume [µL] was calculated by plotting the sum of clearance at all time points over time [min]. The resulting slope PS [µL/min] was calculated inversely. For calculation of the PS_cell_, the inverse PS blanks were subtracted, as shown below in Formula (2).
(2)$$\frac{1}{PS_{cell}}=\frac{1}{PS_{all}}-\frac{1}{PS_{blank}}$$

The final permeability coefficient is calculated by dividing the inverse PS_cell_ by the area of the ThinCerts (0.336 cm^2^) and multiplying by a conversion factor to give the permeability coefficient as µm/min.

### 2.5. Inflammation Study

ThinCerts^®^ were seeded with cells as described above. Stock solutions of tumour necrosis factor α (TNF-α, Sigma-Alrich, H8916, 10 µg), Interleukin-1ß (IL-1ß, PeproTech, 200-01B, 2 µg) and Interferon-γ (IFN-γ, PeproTech, Vienna, Austria, 300-02, 20 µg) were prepared with Dulbecco’s Phosphate Buffered Saline (DPBS, Thermo-Fisher Scientific, Vienna, Austria, 14190094) containing 0.1% bovine serum albumin (BSA, Roth, Graz, Austria, 8076.2) at a concentration of 10 µg/mL and stored at −20 °C until usage. Three days before the experiments, cell layers and blanks were washed twice on the basolateral compartment with 900 µL cultivation media (DMEM supplemented with 10% Fetal Calf Serum and 1% Penicillin/Streptomycin) to remove anti-inflammatory hydrocortisone residues (a component of media supplement HKGS) and were further cultivated in cultivation media. Solutions for the inflammation study were prepared on the day of experiment in DMEM containing 3% 0.1%BSA/DPBS and 3% sterilised dd-H_2_O (control), 100 ng/mL of TNF-α, IL-1ß and IFN-γ each in 3% 0.1%BSA/DPBS and 3% sterilised dd-H_2_O (INF), 100 ng/mL of TNF-α, IL-1ß and IFN-γ each in 3% 0.1%BSA/DPBS and 100 µM ibuprofen (INF + Ibu) or 3% 0.1%BSA/DPBS and 100 µM ibuprofen (Ibu).

Prior to treating cells with prepared DMEM solutions, TEER was measured in cultivation media DMEM supplemented with 1% Penicillin/Streptomycin and 10% Fetal Calf Serum. Subsequently, cell layers and blanks were washed with DMEM (without supplements) twice on the apical and basolateral compartment, followed by TEER measurements in DMEM. Media of cells were replaced with previously prepared DMEM solutions. Control media was applied on the blanks. After 48 h treatment at 37 °C, 5% CO_2_ and 95% humidity TEER was measured. Subsequently, 10 µM carboxyfluorescein was added to the apical DMEM solutions and incubated in the incubator at 37 °C for 2 h. Media were immediately collected and measured for the RFU content using the EnSpire as described above. Afterwards, media were stored at −20 °C until further usage. Two cell-grown ThinCerts^TM^ of each treatment were pooled as one sample with 350 µL lysis buffer supplemented with 1% ß-mercaptoethanol and stored at −80 °C for RNA isolation.

Transport studies with ibuprofen were performed directly after permeability studies with carboxyfluorescein. Selected cells and blanks were pre-incubated with DMEM containing 100 µM probenecid for 30 min at 37 °C, as described above. Apical media were replaced with DMEM solutions containing 100 µM ibuprofen, 100 µM diazepam and 10 µM carboxyfluorescein. Solutions for cells and blanks pre-incubated with probenecid additionally contained 100 µM probenecid on the apical and basolateral compartment. Solutions for cells and blanks without prior probenecid treatment contained 1% sterile DMSO as vehicle instead. Transport studies were subsequently performed, as described above in Section 2.2.

### 2.6. Quantitative Real-Time PCR

Cell samples were lysed, and RNA was isolated and transcribed to cDNA synthesis, as described in detail previously [24]. In short, RNA of cell lysates was isolated using the NucleoSpin RNA kit (Machery Nagel, 740961) and 1 µg RNA was transcribed in cDNA using the High-Capacity Reverse Transcriptase Kit (Applied Biosystems^TM^ Thermo Scientific, 4374967) synthesis. For qPCR, samples were analysed as triplicates using the LightCycler480 II (Roche) with the program described in Lin et al. [24] with primers for 18SrRNA (5′-3′ forward: ATGGTTCCTTTGGTCGCTCG, 5′-3′ reverse: GAGCTCACCGGGTTGGTTTT), Janus Kinase 1 transcript variant a (JAK1tva) (5′-3′ forward: TGACCGTCACCTGCTTTGAG, 5′-3′ reverse: GGTTGGAGATTTCTCGGGGC), JAK1tvb (5′-3′ forward: GGGATATTTCCCTGGCCTTCT, 5′-3′ reverse: AAGAGATCCAGAGGACCCCC), JAK1tvc (5′-3′ forward: CTTTGCCCTGTATGACGAGAAC, 5′-3′ reverse: ACCTCATCCGGTAGTGGAGC), TNF Receptor Associated Factor 2 (TRAF2) tvb (5′-3′ forward: AAAGCAGTTCGGCCTTCCC, 5′-3′ reverse: TCCTTTTCACCAAGGCGGAC), TRAF2tvc (5′-3′ forward: CCTTCCCAGATAATGCTGCCC, 5′-3′ reverse: GCTCTCGTATTCTTTCAGGGTC) and nuclear factor κB (NF-κB) RelA (forward 5′-3′: ACTGTTCCCCCTCATCTTCC, reverse 5′-3′: TGGTCCTGTGTAGCCATTGA). Data were acquired with the LightCycler480 V1.5 software. ΔCt values were calculated by subtracting Ct values of 18SrRNA. Values were negatively exponentiated to the power of 2 and normalised to samples treated with inflammatory cytokines.

### 2.7. Western Blot

Cells seeded on 6-well ThinCerts for protein analysis were cultivated as described in the method in Section 2.1 using 2 mL media on the apical and 3.5 mL on the basolateral side. On day 29, cells were treated with DMEM containing 3% 0.1%BSA/DPBS plus 3% sterile dd-H_2_O (control), 100 ng/mL cytokines plus 3% sterile dd-H_2_O (INF), 100 ng/mL cytokines plus 100 µM ibuprofen (INF+IBU) as well as 3% 0.1%BSA/DPBS plus 100 µM ibuprofen (IBU) for 48 h, as described in the method in Section 2.5. Media were collected and stored at −20 °C, while cell layers were washed twice with pre-cooled DPBS on ice. After 5 min incubation with DPBS after the last washing step, cells were lysed with 50 µL RIPA buffer (50 mM TRIS pH 8.0; 150 mM NaCl, 0.1% SDS, 0.5% sodium-deoxycholate, 1% NP40), supplemented with complete ULTRA protease inhibitor cocktail and PhosphoSTOP minitablet (Roche Applied Science, Penzberg, Germany, 288 05892970001 and 04906837001) for 30 min. Cell lysates were stored at −80 °C and protein concentration was determined with the Pierce BCA assay kit (Thermo Fisher, Vienna, Austria,23227) according to the manufacturer´s instruction. Western blotting was performed as described previously [24]. In short, 20 μg protein was loaded from each sample and primary antibody for COX-1 (Santa Cruz, Dallas, TX, USA, sc-19998, mouse) and COX-2 (Santa Cruz, sc-19999) was applied 1:200 and 1:200 in 5% non-fat dried milk (AppliChem Panreac, Darmstadt, Germany, A0830,0500). Subsequently, the HRP-labelled anti-mouse antibody was applied 1:5000 diluted in 5% non-fat dried milk (Cell signalling Technology, 7076S). For visualization of endogenous control ß-actin, HRP-labeled ß-actin antibody (Sigma-Aldrich, A3854, 7076S) was applied 1:20,000 in 5% non-fat dried milk. Signals were captured with Chemi-Doc Imaging system (BioRad, Hercules, CA, USA) and analysed with ImageLab Software Version 5.2.1.

### 2.8. Quantibody

Media samples from inflammation studies on 24-well inserts were collected at the end of the experiment after 48 h, stored at −20 °C until analysis and thawed on ice on the day of quantification. Applied stock solution, apical and basolateral media from blanks and cells of all treatment conditions were used for measurements and diluted 1:2 in reagent diluent. Samples were processed according to the manufacturer’s instructions of the Quantibody^®^ Human Inflammation Array (Q1, QAH-INF-1-4, RayBiotech, Peachtree Corner, GA, USA) to quantify CCL2 (MCP1), CXCL8 (IL-8), INF-γ, IL-10, IL-13, IL-1α, IL-1β, IL-4, IL-6 and TNFα. Slides were recorded with an array scanner, images of the scanned arrays were processed by means of grid analysis and quantified with an xls-template from RayBiotech specified for the obtained array.

### 2.9. Statistical Analysis

Results are shown as mean ± SD and mean ± SEM when equipped for graphs with large sample sizes. Graphs were illustrated with SigmaPlot version 14.0 and statistical analysis was performed as one-way or two-way ANOVA with post hoc Holm–Sidak test and * *p* < 0.05, ** *p* < 0.01, *** *p* < 0.001 and α = 0.05 using SigmaPlot.

## 3. Results

### 3.1. Permeability of NSAIDs across the Oral Mucosa In Vitro

Figure 1A–D shows the cleared volume curves over time for the transport studies of NSAIDs and diazepam across blank and cell layer inserts. The black curves of the compounds across the blank inserts without cells indicate faster transport in comparison to the corresponding inserts with the cells, which confirmed that the oral mucosa barrier in vitro model formed a distinct barrier for the tested NSAIDs. In addition, it was recognizable that the blank curves for diazepam and diclofenac, ibuprofen and piroxicam were very similar, underlining also very similar interactions of these compounds with the blank membranes. Notably, lower amounts of celecoxib already permeated across the blank inserts than for diazepam, indicating more interaction of celecoxib with the membrane and highlighting the importance of blank studies across only the membranes for each compound in order to account for these differences when assessing permeability coefficients. The white cleared volume curves over time suggested that diazepam permeated faster than every NSAID, which was expected for the transcellular marker diazepam. Notably, the paracellular marker substance carboxyfluorescein (CF) showed the highest clearance across blanks and cells of all substances applied (Figure 1E).

Corresponding to the cleared volume curves, the calculated permeability coefficients of paracellular marker CF (4.63 ± 0.24 µm/min, *p* < 0.001) were significantly lower than for the transcellular marker diazepam (11.90 ± 1.33 µm/min) (Figure 1F). For applied NSAIDs, ibuprofen showed the highest permeability coefficient (5.77 ± 0.60 µm/min, not significant (n.s.) compared to diazepam), followed by piroxicam (4.86 ± 0.84 µm/min, *p* < 0.05) and diclofenac (3.64 ± 0.34 µm/min, *p* < 0.05). Of all NSAIDs, celecoxib showed the lowest permeability coefficient (0.44 ± 0.10 µm/min, *p* < 0.01). However, upon addition of verapamil or probenecid, celecoxib displayed the highest increase of 4.72-fold (2.10 ± 0.92 µm/min, *p* < 0.01 vs. diazepam+verapamil) with verapamil and a 13.63-fold (12.74 ± 1.31 µm/min, n.s.) increase with probenecid. On the other hand, the permeability coefficient of diclofenac did not change after verapamil treatment (3.54 ± 0.38 µm/min, *p* < 0.01 vs. diazepam+verapamil.), while the addition of probenecid led to a 1.83-fold higher permeability coefficient (6.66 ± 1.87 µm/min, n.s.). For ibuprofen and piroxicam, the addition of verapamil showed a slight increase of 1.23-fold for both NSAIDs (ibuprofen: 7.13 ± 0.51 µm/min, n.s.; piroxicam: 6.00 ± 0.70 µm/min, *p* < 0.05 vs. diazepam + verapamil) compared to no inhibitor. On the other hand, added probenecid led to a 1.36-fold increase in ibuprofen permeation (7.87 ± 1.32 µm/min, n.s.) and to a 1.19-fold increase in the permeability coefficient of piroxicam (5.81 ± 0.61 µm/min, n.s.).

As additional analysis, permeability coefficients of NSAIDs and diazepam were normalised to corresponding permeability coefficients of paracellular marker CF, shown in Figure 1E. As a result, diazepam showed very similar permeability coefficients regardless of the treatments (control: 2.65 (±0.27)-fold, added verapamil: 2.72 (±0.61)-fold, added probenecid: 2.51 (±0.25)-fold in comparison to CF). Similarly as described above, celecoxib showed the lowest permeability coefficient (0.12 ± 0.04-fold, *p* < 0.05 vs. diazepam), which was increased by verapamil (0.38 ± 0.15-fold, *p* < 0.05 vs. diazepam + verapamil) and probenecid (1.29 ± 0.42-fold) treatments, followed by diclofenac (0.77 ± 0.08-fold, +verapamil: 0.86 ± 0.08-fold, +probenecid: 1.10 ± 0.21-fold). Similarly, ibuprofen showed the highest permeability (1.41 ± 0.40-fold) of all NSAIDs, increased by verapamil (1.70 ± 0.46-fold) and probenecid (1.51 ± 0.40-fold) addition. Piroxicam displayed a permeability coefficient of 1.06 ± 0.14-fold without inhibitors, and a permeability coefficient of 1.38 ± 0.16 after addition of verapamil and 1.20 ± 0.19-fold under probenecid (an overview of calculated slope values for calculation of the permeability of cells and blanks with corresponding TEER values is shown in Appendix A).

### 3.2. Cytokines Induce Inflammatory Response in the Oral Mucosa Epithelium In Vitro Model

NSAIDs are very often applied during diseases to suppress inflammatory processes. It is known that inflammation can change the functionality of biological barriers, in particular transporter proteins as well as tight junctions determining the transcellular as well as the paracellular route of compounds. As a next step, we wanted to know whether the transport of NSAIDs would be changed under inflammatory conditions. Before conducting transport studies with NSAIDs across our inflamed oral mucosa in vitro model, we assessed effects of the applied cytokine mixture on our model to confirm that the stimuli induced inflammatory changes on the mRNA and the protein level. For these studies, we decided to investigate the effects of ibuprofen, because ibuprofen belongs to the most frequently used NSAIDs in Europe, Asia and Australia next to diclofenac [30,31], and ibuprofen has a bigger potential for future use, since a prevalence for stroke was found for diclofenac but not for ibuprofen [32]. First, we analysed the expression of JAK1, TRAF2 and RELA at the mRNA level by qPCR after cytokine treatments. All transcript variants of JAK1 and TRAF2 were upregulated after cytokine addition in comparison to control cells as well as cells treated with ibuprofen only, but the addition of ibuprofen during inflammation did not block the upregulation of JAK1 and TRAF2 (Figure 2A). In the case of RELA, ibuprofen was also able to inhibit the increase in RELA by inflammatory cytokines in a significant manner (*p* < 0.05), confirming the anti-inflammatory properties of ibuprofen in our model.

At the protein level, the expression of COX-1 and COX-2 was verified by Western blotting using samples of inflammation studies. While all treatment groups showed a similar expression of COX-1, an increased expression of COX-2 was observed for samples treated with cytokines or cytokines plus ibuprofen (Figure 2B). Moreover, it was tested whether the inflammatory stimuli also led to the secretion of cytokines. Concentrations of cytokines were measured in media of the apical (saliva) and basolateral (blood) compartments after cytokine mixture treatment. In the apical media, higher levels of IL-1 α, IL-1ß, IL-6, MCP-1 and TNF-α were detected for samples treated with cytokines or cytokines with ibuprofen. IFN-γ levels of samples treated with cytokines were significantly higher (*p* < 0.01) compared to samples treated with ibuprofen or control samples. No regulation was observed for IL-8 among the treatment groups. In media from the basolateral compartment, higher levels of IL-1 α, IL-1ß, IL-6, MCP-1, IFN-γ and TNF-α were observed in samples treated with cytokines or cytokines and ibuprofen, while lower or no levels were observed for control samples or samples treated with ibuprofen. IL-4, IL-10 and IL-13 were not detected with the applied Quantibody^®^ array after 48 h stimulation. In summary, inflammation of the oral mucosa epithelium in vitro model was confirmed at the mRNA and protein level by increase in respective marker molecules and the secretion of cytokines. After the inflammation protocol for the model was successfully established, the effects of the inflammation on transport routes were investigated.

### 3.3. Affected Transport Routes after Treatment with Pro-Inflammatory Cytokines

First, it was tested whether the inflammation affected the paracellular barrier integrity of the model. For this, transepithelial electrical resistance (TEER) was measured at the start of the experiment and after treatment with cytokines. Due to reproducibility reasons, the threshold for cell layers to be included into the study was set to a minimum TEER value of at least 100 Ω × cm^2^. TEER values at start and after 48 h are shown in Figure 3A. While control cells and cells treated with ibuprofen showed similar TEER values after 48 h (77.10 ± 15.25 Ω × cm^2^, 69.65 ± 9.10 Ω × cm^2^), treatment with cytokines or cytokines plus ibuprofen displayed significantly lower (*p* < 0.001) TEER values of 45.30 ± 5.10 Ω × cm^2^ and 49.16 ± 14.51 Ω*cm^2^, respectively. Permeability assays performed with CF for two hours after 48 h inflammation treatment revealed a lower permeability coefficient for CF across control cells (12.41 ± 5.15 µm/min) and cells treated with ibuprofen (12.83 ± 3.57 µm/min) in comparison to cells treated with cytokines (16.16 ± 13.05 µm/min) or cytokines with ibuprofen (16.75 ± 9.76 µm/min), shown in Figure 3B. Thus, CF permeability data confirmed TEER data that inflammatory stimuli reduced the model’s paracellular tightness and that the addition of ibuprofen did not block the cytokine-induced barrier breakdown.

To evaluate the effects of inflammation on transcellular transport routes, transport studies with ibuprofen plus probenecid were carried out. Ibuprofen and probenecid were chosen, since previous studies showed that ibuprofen was a substrate of transporters whose activity was inhibitable by probenecid [33]. As described above, CF and diazepam were added as control compounds during transport studies and permeability coefficients were calculated, drawing samples every 60 min for 4 h. For control cells and cells treated with ibuprofen, permeability coefficients of CF were quite similar (control: 14.26 ± 7.14 µm/min, plus ibuprofen: 14.39 ± 3.12 µm/min), whereas exposure to the cytokine cocktail with and without ibuprofen led to an increased permeability of CF (cytokine: 24.30 ± 4.82 µm/min, cytokine + ibuprofen: 25.04 ± 6.73 µm/min). This cytokine treatment dependency was also observed for permeability coefficients of ibuprofen, as the pre-treatment with cytokines or cytokines plus ibuprofen led to higher permeability coefficients (cytokine: 20.01 ± 12.92 µm/min; cytokine + ibuprofen 21.06 ± 14.25 µm/min) than across control cell layers (14.38 ± 7.25 µm/min) or cell layers treated with ibuprofen alone (11.55 ± 1.23 µm/min).

The addition of probenecid showed the tendency to decrease permeability coefficients of CF. For control cell layers and cell layers treated with ibuprofen, a reduction of 14.36% and 24.59% (not significant) was measured, while for cells treated with cytokines or cytokines and ibuprofen, a reduction of 25.03% and 18.90% was observed (not significant). On the contrary, the addition of probenecid showed the tendency of an increased permeability of ibuprofen. In the control group, the addition of probenecid led to a 42.46% higher permeability of ibuprofen (*p* < 0.001 vs. diazepam) and a 52.64% (not significant) higher permeability for cells treated with ibuprofen previously. However, cells treated with cytokines showed no distinct regulation of ibuprofen permeability with or without probenecid. In summary, inflammation increased permeation of CF and ibuprofen, which was probably due to a significantly decreased paracellular tightness. Interestingly, the addition of probenecid did not increase CF permeability, but enhanced ibuprofen permeability only across non-inflamed cell layers in the same experimental setting. This indicated that the probenecid-inhibitable part of ibuprofen transport was already affected after inflammation and that ibuprofen transport could not further be increased.

## 4. Discussion

Even though up-to date NSAIDs are commonly prescribed orally, buccal delivery offers the major advantage of by-passing the gastrointestinal tract. The latter limits the adverse side effects caused by long-term usage of NSAIDs, such as gastric erosions or ulcers. Recently, technical advances were made to optimise buccal formulations for NSAIDs. For instance, Shirvan et al. (2021) developed a patch with potential for therapeutic usage by electrospinning of chitosan, polyvinylalcohol and ibuprofen and Eleftheriadis et al. (2020) described a printing technique with ibuprofen for personalised mucoadhesive films [34,35]. While an optimised formulation of patches or films is crucial to ascertain the delivery of drugs of interest at therapeutic concentrations, a thorough understanding of the transport mechanism of drugs of interest is essential to improve their buccal delivery.

For this purpose, we investigated the buccal transport of NSAIDs, using an established multi-layered model of the buccal mucosa to determine the permeability of celecoxib, diclofenac, ibuprofen and piroxicam. Diazepam, a standard for the transcellular transport route, was used for comparison [33]. Of applied NSAIDs, ibuprofen showed the highest permeability, followed by piroxicam, diclofenac and celecoxib (Figure 1F). Corresponding to this, literature data of a monolayer epithelial model of the humane intestine (Caco-2) showed the same ranking for NSAIDs, demonstrating the highest permeability for ibuprofen, followed by piroxicam [36], diclofenac and celecoxib [37,38]. In contrast, transport studies with monolayer brain endothelial models based on porcine (PBMEC/C1-2) and human (ECV304) cell lines showed the highest permeability for piroxicam, followed by ibuprofen, celecoxib and diclofenac for PBMEC/C1-2 or celecoxib and diclofenac for ECV304 [33]. Nevertheless, similar to our model, diazepam showed a higher permeation rate than NSAIDs in all three models (Caco-2, PBMEC/C1-2 and ECV304) [33,36]. Interestingly, the permeability coefficients of paracellular marker carboxyfluorescein (CF) across ECV304 cell monolayers were very similar compared to our model (~5 µm/min), while the permeability coefficients for the transcellular marker diazepam were distinctly higher across the ECV304 model (~20–40 µm/min) than in our model (~12 µm/min), reflecting the influence of multilayers (as in our buccal epithelium model) on the permeation of substances such as diazepam [33].

In accordance with the permeability coefficient, after four hours, the highest cumulated concentration of NSAID in the basolateral compartment was found for ibuprofen (0.067 ± 0.0078 nmol/mm^2^, mean ± SD, *n* = 6), followed by piroxicam (0.055 ± 0.016 nmol/mm^2^, mean ± SD, *n* = 8), diclofenac (0.043 ± 0.0046 nmol/mm^2^, mean ± SD, *n* = 6) and celecoxib (0.0071 ± 0.00032 nmol/mm^2^, mean ± SD, *n* = 4).

Since all NSAIDs penetrated slower than transcellular marker diazepam, which is proposed to permeate by passive diffusion, the question arose about the role of transporter proteins for NSAID transport. While NSAIDs are known for their inhibitory activity on cyclo-oxygenase enzymes (COX) affecting prostaglandin synthesis, they have also been described to have a direct inhibitory effect on transporters of prostaglandins. The uptake of prostaglandins is mainly facilitated by organic anion transporters (OATs) [39], while their efflux is regulated by the multidrug resistance protein 4 (MRP4), an ATP-binding cassette (ABC) transporter, associated with multidrug resistance [40]. Ibuprofen was able to inhibit transporter MRP4 (ABCC4) in human embryonic kidney cells (HEK293) at a concentration of 200 µM [40] or in peripheral blood lymphocyte (PBL) cells at a concentration of 10 µM. In the latter case, ibuprofen treatment increased the intracellular levels of the antiretroviral drug zidovudine, whose efflux is facilitated by MRP4 [41]. The inhibitory effects of diclofenac, celecoxib and piroxicam on MRP4 function in HEK203 cells were weaker in comparison to ibuprofen [40,42]. These data confirmed interactions of the investigated NSAIDs with transporter proteins. Thus, we decided to test the influence of inhibitors of multidrug resistance transporters, probenecid and verapamil, on the transport properties of NSAIDs in our model. Probenecid is known to inhibit MRP4 [43] as well as OAT1 or OAT3 [44], while verapamil is a well-known inhibitor of P-glycoprotein (ABCB1) as well as of MRP1 (ABCC1) [45]. Results revealed overall a higher inhibitory effect by probenecid on NSAID transport than verapamil, increasing the permeability of celecoxib, diclofenac and ibuprofen, whereas an increase upon verapamil was only detected for piroxicam. As the addition of multidrug resistance transporter inhibitors affected the transport of the applied NSAIDs, a concentration-dependent, active transport component could be assumed. The presence of a concentration-dependent permeation behaviour of NSAIDs across cell membranes, especially ibuprofen, has been described in detail previously [46]. A very slight increase in the permeability of paracellular marker CF was detected after probenecid addition, which could be related to publications reporting that CF might be also a moderate substrate for OAT [47] or MRP [48] transporters.

A literature search for in vivo or ex vivo studies with free NSAID led only to limited results. For example, permeation of piroxicam in combination with various cyclodextrins for better solubility was investigated by Kontogiannidou et al. (2019) ex vivo across porcine oral mucosa, who reported an apparent permeability, for instance, for a 1:1 mixture of piroxicam:ß-cyclodextrin of 0.21 ± 0.08 × 10^−3^ cm/h (*n* = 5, mean ± SD), equivalent to 0.035 ± 0.013 µm/min [49]. This permeation coefficient for piroxicam:ß-cyclodextrin across porcine oral mucosa was over 80 times smaller than the result shown across our in vitro model. This could be due to (i) the significantly higher thickness of the porcine mucosa (600–850 µm) compared to our cellular model and (ii) the lack of blank studies considering the effects of the tissues’ extracellular matrix for the calculation of permeability coefficients in ex vivo studies. Moreover, the addition of solubility-enhancing components such as ß-cyclodextrin is common for ex vivo permeability studies, but should also be considered when comparing these data to the presented in vitro results here. In general, testing of whole dosage forms, for instance, often accomplished with ex vivo tissues in the Franz Cell system, should also be feasible with our oral mucosa model.

To evaluate if physicochemical properties of the NSAIDs correlate with their calculated permeability coefficient, publicly available data were used for comparison (see Appendix A). While XlogP3 values showed no correlation with the permeability coefficient (PC) values, CXlogD at pH 7.4 correlated with the permeability coefficient values with R^2^ = 0.5797 (R = 0.761). Calculation of CXlogD at pH 7.4 vs. XlogPC led to a higher R^2^ value of 0.6649 (R = 0.815). Considering the pH value of the experimental set-up improved the correlation, but it should be noted that the found involvement of active transport systems already supported the assumption that lipophilicity and distribution parameters are not sufficient for an accurate prediction of NSAID permeability across the applied oral mucosa in vitro model.

Upon determining the transport ranking of celecoxib, diclofenac, ibuprofen and piroxicam and evidence of involvement of multidrug resistance transporters, we aimed to test whether the transport of NSAIDs might change under inflammatory conditions. For this, we chose to investigate the permeability of ibuprofen in combination with probenecid, since literature research showed the highest popularity for ibuprofen and the smallest risk for severe side effects of the mostly prescribed NSAIDs [30,31,32]. The cultivation conditions of our oral mucosa model were adapted prior to exposure to pro-inflammatory cytokines (TNF-α, IL-1ß and IFN-γ) since HKGS, a supplement in cultivation media, contained hydrocortisone, a corticosteroid with anti-inflammatory properties, in particular to IL-1ß-induced inflammation response [50]. Hence, cellular layers were washed with media without HKGS 72 h prior to the start of inflammation studies, to reduce the content of hydrocortisone on cell layers, considering its biological half-life of 8–12 h [51]. After these 72 h cultivations without HKGS, cells were treated either with inflammatory cytokines (TNF-α, IL-1ß and IFN-γ), ibuprofen, or cytokines plus ibuprofen for 48 h. Ibuprofen was applied at 100 µM, a concentration similar to measured plasma concentration (116–142.5 µM) after oral administration of 200 mg [52,53]. First, the response of our in vitro model on the inflammatory cytokine mixture was evaluated on the mRNA level. Regulation of Janus kinase 1 (JAK1), TNF receptor-associated factor 2 (TRAF2) and RelA was determined. While JAK1 is downstream of the INF-γ receptor, involved in signaling of pro-inflammatory cytokines, such as IL-6, IL-7, IL-10 or IL-21, and serves as a therapeutic target in autoimmune diseases, such as bowel disease [54], TRAF2 mediates the TNF-α mediated response and is essential for the activation of transcription factors, such as the nuclear factor-κBs (NFκB) [55]. Even though all transcription variants of JAK1 and TRAF2 showed an upregulation upon exposure to cytokines (Figure 2A), no difference was measured in comparison to cells treated with cytokines plus ibuprofen. On the other hand, RelA (also known as p65), a member of the NF-κB family [56] and involved in regulation of NF-κB down the “canonical” pathway [57], was upregulated by cytokine treatment, which was partly blocked by ibuprofen, confirming the anti-inflammatory effects of ibuprofen on this specific pathway.

Since NSAIDs mainly inhibit the activity of COX-1 and COX-2, with the latter increasingly expressed during inflammation, their regulation was analysed on the protein level (Figure 2B). COX-1 was ubiquitously expressed in all treatment groups, while COX-2 was only expressed in cells treated with cytokines, confirming the pro-inflammatory cytokine-dependent upregulation of COX-2. Interestingly, cells treated with cytokines and ibuprofen showed a distinct higher regulation of COX-2 compared to cells treated with cytokines alone. In this regard, it was shown that NSAIDs could display an inhibitory effect on TNF-induced NF-κB activity, and since NF-κB is a key regulator for expression of COX-2, inhibition of NF-κB leads to suppression of COX-2 expression [58]. Besides this, *Paik* et al. (2000) demonstrated an NF-κB-independent upregulation of COX-2 in human cell lines by an NSAID (flufenamic acid) [59]. Moreover, upregulation of COX-2 was found in prostate cancer cells (PC3 cells) after treatment with 1500 µM ibuprofen for 24 h [60] and a recent study by Chai et al. (2015) demonstrated similar findings [61]. However, no up-regulation of COX-2 by ibuprofen was found in our cells without the addition of cytokines, suggesting an induced COX-2 expression boosted by the combination of pro-inflammatory cytokines with ibuprofen after 48 h treatment. In this context, clinical studies reported induced COX-2 expression in the oral mucosa 48 h post-surgery in patients treated with ibuprofen and rofecoxib compared to the placebo group [62]. In addition, higher levels of COX-2 were found after treatment with NSAIDs post-surgery [63].

Next to the increased COX-2 expression levels in the cells of our oral mucosa model after the addition of cytokines, we measured levels of cytokines (IL-1α, IL-1ß, IL-6, IL-8, IFN-γ, TNF-α) and chemokine MCP-1 in the apical and basolateral compartments. After 48 h treatment, levels of added cytokines IL-1ß, IFN-γ and TNF-α were still distinctly higher in the media of cells which were treated with these cytokines in comparison to cells which did not receive the cytokine mixture. In this regard, it was not possible to define how much the residual amount of non-degraded cytokines contributed to the detected concentration of cytokines. However, concentrations of non-added cytokines IL-1α and IL-6 and chemokine MCP-1 were also distinctly higher in the media from cells treated with the cytokine mixture, confirming that these cell layers were inflamed and secreted signaling molecules into the media. In general, apical concentrations of cytokines IL-1α and IL-6 and chemokine MCP-1 were higher than the concentrations found in the basolateral compartment, indicating that the secretion was probably restricted to the blood compartment. For the leveling of these numbers, the different volumes of the apical and basolateral compartments leading to a different dilution factor per cell surface as well as the possible influence of the porous plastic membranal support restricting the access to the basolateral compartment versus free access to the apical fluid should be considered. In accordance with our model set-up and findings, elevated concentration of IFN-γ was measured in inflamed oral mucosa tissue in comparison to healthy tissue in a clinical trial [64]. Interestingly, we found very minor regulation of IL-8 in our model. Using the same cell line as we did, cultivated on 24-well plates for one week, Tetyczka et al. (2021) showed an elevated protein level of IL-8 in the supernatants of media after 24 h exposure to IL-1ß and TNF-α (100–400 ng/mL) [65]. While IFN-γ is described to be involved in inflammatory responses and cell death, a recent study using knock-out mice reported aggravated inflammatory response due to deficiency of IFN-γ [66]. Previous studies reported similar results, suggesting anti-inflammatory activities of IFN-γ [67,68]. In this regard, it could be speculated that the additionally applied IFN-γ in our experiments might be responsible for the absence of regulation of IL-8 in our model. However, preliminary studies with our model showed no significant disruption of the integrity of the paracellular barrier upon exposure to IL-1ß and TNF-α alone, in contrast to our findings shown in Figure 3, suggesting a direct link of IFN-γ to the disruption of the paracellular barrier. This was confirmed by previous studies describing a reversible defect on the paracellular barrier of an epithelial intestine model upon exposure to IFN-γ [69].

Moreover, media without serum led to lower TEER overall after 48h (Figure 3). A significant reduction was observed for cells treated with cytokines or cytokines with ibuprofen compared to controls, possibly linked to higher permeability of carboxyfluorescein and ibuprofen in those treatment groups. Interestingly, while probenecid was shown to increase the permeability of ibuprofen in controls, similarly to previous transport studies, no tendency of increase was observed in cells treated with cytokines. This indicated a shift of efflux and influx transporters upon treatment with cytokines, either through downregulated efflux systems or possibly through increased inhibition of influx transporters by probenecid. In this regard, 48 h cytokine treatment of our oral mucosa in vitro model led to significant regulation at the mRNA level of transporter and tight junction proteins such as ABCC4 (MRP4), determined by high-throughput qPCR (see Appendix A). Similarly, previous studies have shown a downregulation of efflux transporter ABCG2 at the mRNA level and a time-dependent (6, 24, 48 h) regulation of protein expression for ABCB1 and ABCG2 upon exposure to IL-1ß and TNF alpha at the blood–brain barrier [70]. Characterisation of the used in vitro model based on TR146 cells demonstrated the expression and functionality of those efflux transporters [29]. Hence, future studies should include analysis of the expression of ATP-binding cassettes as well as solute carriers on the protein and functional level at several time points after inflammation to determine the regulatory mechanism of transporters involved upon inflammation stimuli. Additionally, since administration of multidrug resistance transporter inhibitors suggested the presence of active transport systems for the investigated NSAIDs in the TR146 model from the apical to the basolateral compartment, future studies could include transport studies from the basolateral compartment to the apical compartment to analyse a potential polarisation of NSAID transport, as previously detected for C-reactive protein in the TR146 model [24].

## 5. Conclusions

All in all, we applied a model of the oral mucosa epithelium for transport studies with various NSAIDs in combination with inhibitors for active transporter systems, showing distinct substance-dependent differences, with the highest permeability for ibuprofen. Additionally, the buccal mucosa epithelium model showed an inflammatory response after treatment with pro-inflammatory cytokines at the mRNA and protein level and revealed an impaired paracellular barrier linked to IFN-γ. Transport studies demonstrated an elevated transport of ibuprofen across inflamed oral mucosa epithelial cells but no increased permeability of ibuprofen with probenecid in contrast to controls, suggesting a shift of transporter systems upon exposure to inflammatory stimuli.

## Figures and Tables

**Figure 1 pharmaceutics-16-00543-f001:**
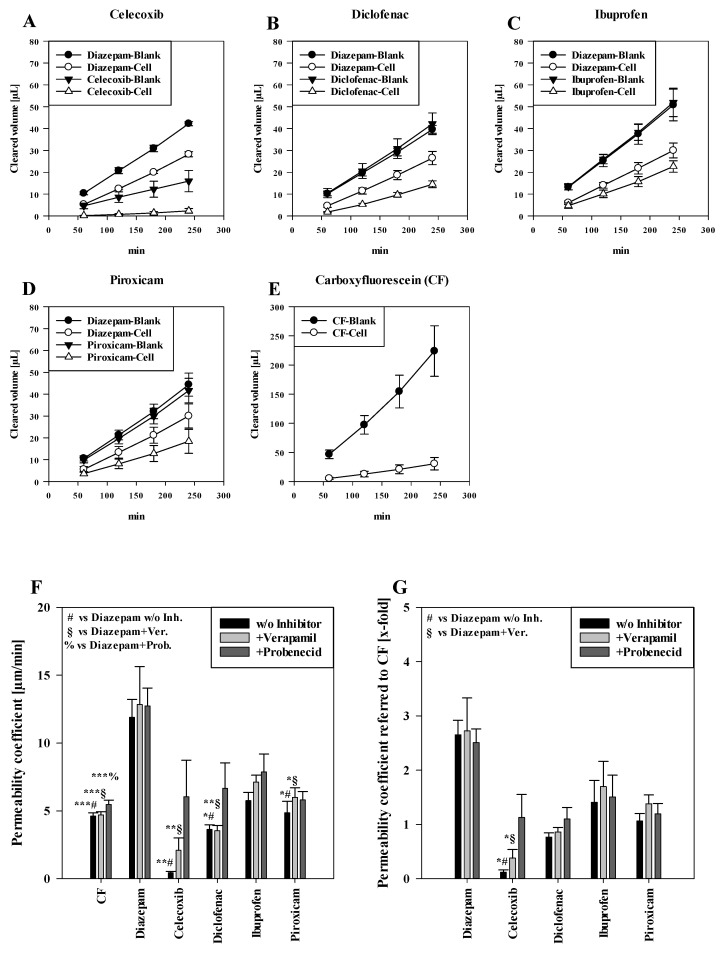
Measured cleared volume [µL] from blanks and cells over time [min] of (**A**) celecoxib, (**B**) diclofenac, (**C**) ibuprofen and (**D**) piroxicam with corresponding diazepam values from experiments. Representative values for carboxyfluorescein (CF) measured in the piroxicam study shown in (**E**). Results shown as mean ± SD from three independent experiments (*n* = 4–8). Permeability coefficient of carboxyfluroescein, diazepam and NSAIDs without inhibitors and upon treatment with verapamil and probenecid shown as µm/min (**F**) and normalised to the respective permeability coefficient of CF (**G**). Results shown as mean ± SEM from three independent experiments (*n* = 4–28). Statistical analysis performed as two-way ANOVA with post hoc Holm–Sidak test, * *p* < 0.05, ** *p* < 0.01, *** *p* < 0.001 and α = 0.05.

**Figure 2 pharmaceutics-16-00543-f002:**
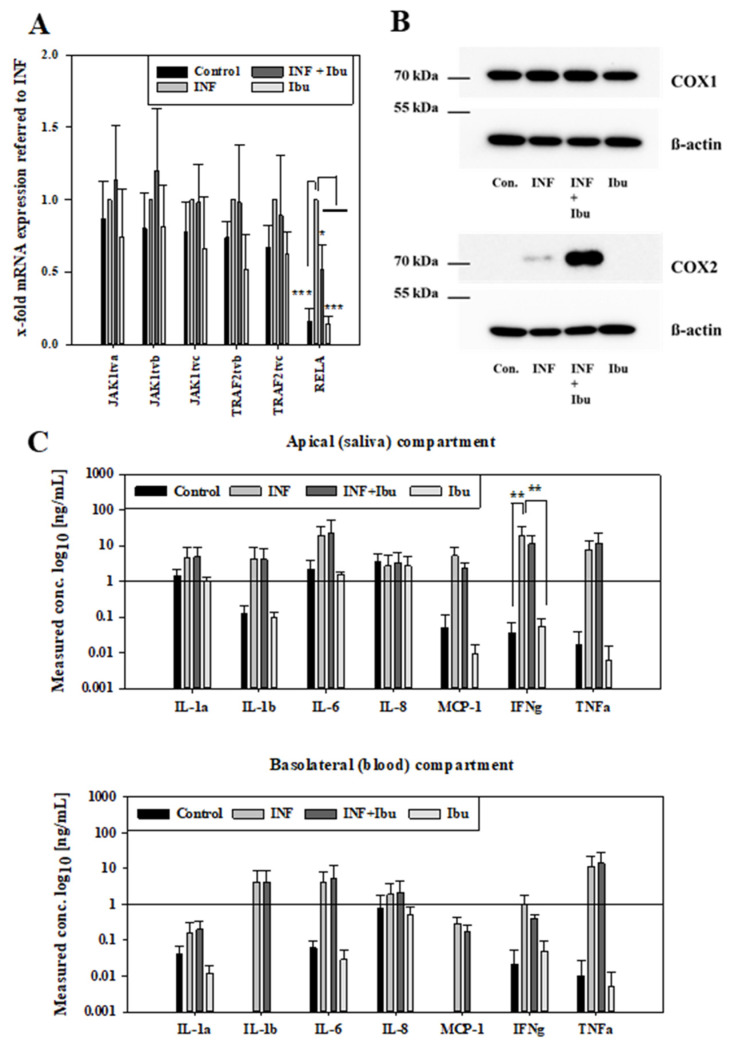
Samples from 48 h inflammation studies (control: media, INF: 100 ng/mL IL-1ß, IFN-γ and TNF-α, INF+Ibu: 100 ng/mL IL-1ß, IFN-γ, TNF-α and 100 µM ibuprofen, Ibu: 100 µM ibuprofen) analysed for mRNA expression of JAK1, TRAF2 and RELA. (**A**) Expression values referred to housekeeping gene 18SrRNA and shown normalised to samples treated with INF as mean ± SD from three independent experiments. Statistical analysis performed as two-way ANOVA with post hoc Holm–Sidak test, * *p* < 0.05, *** *p* < 0.001, α = 0.05. (**B**) Representative Western blot for COX-1 (70 kDa), COX-2 (70–72 kDa) and ß-actin (42 kDa) after inflammation studies. (**C**) Measured concentration of cytokines in media from inflammation studies on the apical (saliva) and basolateral (blood) compartment. Results shown as mean ± SD from three independent experiments. Statistical analysis performed as two-way ANOVA with post hoc Holm–Sidak test, ** *p* < 0.01 and α = 0.05.

**Figure 3 pharmaceutics-16-00543-f003:**
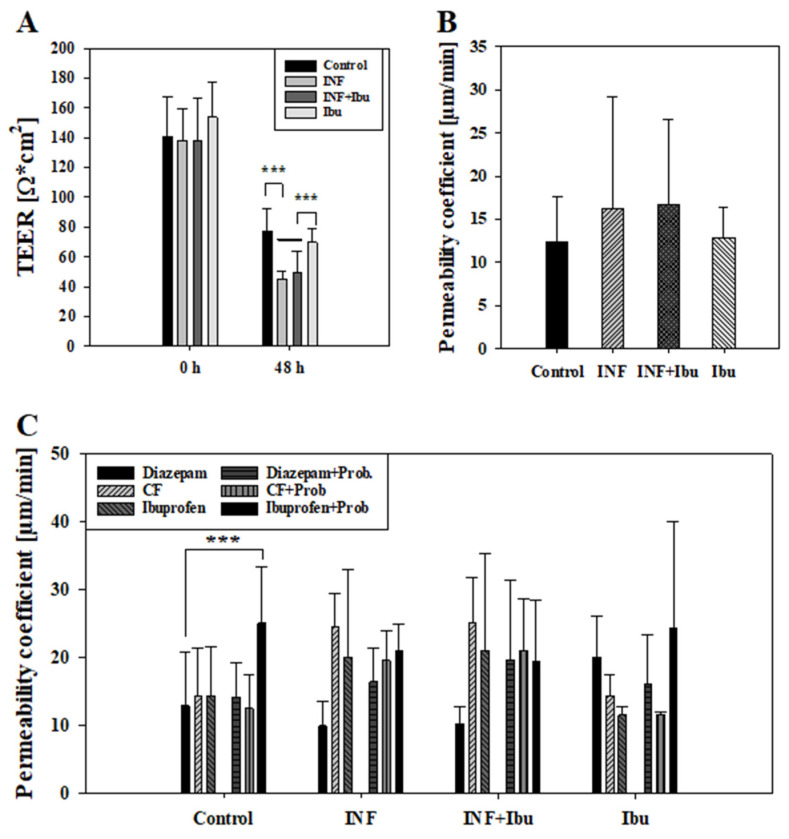
Measured TEER values [Ω × cm^2^] at start (0 h) and end (48 h) of experiment from control cells, cells treated with cytokines (INF), cytokines and 100 µM ibuprofen (INF + Ibu) and 100 µM Ibuprofen (Ibu) (**A**). Cells showing TEER values lower than 100 Ω*cm^2^ at beginning of experiment were excluded from further data analysis. Results shown as mean ± SD from three independent experiments (*n* = 13–17). Statistical analysis performed as one-way ANOVA following post hoc Holm–Sidak test with *** *p* < 0.001 and α = 0.05. (**B**) Corresponding permeability coefficient values of permeability assays with carboxyfluorescein performed at end of experiment (48 h). Results shown as mean ± SD from three independent experiments (*n* = 13–17). Statistical analysis performed as one-way ANOVA with α = 0.05. (**C**) Measured permeability coefficient from transport studies with diazepam, ibuprofen and carboxyfluorescein (CF) with and without probenecid (Prob) performed after 48 h. Results shown as mean ± SD from three independent experiments (*n* = 3–6). Statistical analysis performed as two-way ANOVA following post hoc Holm–Sidak test with *** *p* < 0.001 and α = 0.05.

## Data Availability

The data presented in this study are available on request from the corresponding author.

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
