# Peer review of "Transport of Non-Steroidal Anti-Inflammatory Drugs across an Oral Mucosa Epithelium In Vitro Model"

_pharmaceutics, 2024, doi:10.3390/pharmaceutics16040543_

Round 1

Reviewer 1 Report

Comments and Suggestions for Authors

The manuscript titled “Transport of non-steroidal anti-inflammatory drugs across an oral mucosa epithelium in vitro model” is somewhat interesting. However, the NSAIDs studied in this research study are not novel molecules and have been extensively studied in the past across other membranes and cells. Below are some concerns with the current manuscript.

It is not clear on the rationale and therapeutic usage of these specific NSAIDs on the buccal delivery.

There is no discussion of the concentration dependency on the transport of ibuprofen, piroxicam, diclofenac, celecoxib across oral mucosa epithelium in vitro model employed in this study.

No ex vivo or in vivo correlation of the transport dynamics of the respective NSAIDs was presented in this study.

Overall, there is no novelty in the research and/or does not significantly influence the field with the current findings.

Reviewer 2 Report

Comments and Suggestions for Authors

Dear Authors,

Your article is very interesting and provides new insights into the development of buccal dosage forms and their in vitro assessment. There are a few points that require further explanation and potential development in your manuscript.

1. The most critical aspect to address is whether different concentrations of compounds were tested. Understanding or at least discussing the impact of concentration on permeability and absorption is essential. Is it possible to establish a linearity range for the conducted test? Can the authors calculate permeability factors by categorizing results into ranges based on the concentration in the donor compartment?

2. Additionally, was an assessment made of the reverse effect - permeability from the acceptor chamber back to the donor chamber? Is this effect asymmetrical or symmetrical?

3. Could you determine the drug dosage per mm² that could be effectively delivered?

4. Why were substances only dissolved in DMSO/ddH2O? Are the authors aware of the potential differences if standard buffers at pH 4-6.5 were used? There is a clear need for development of new tests that bring us closer to the real assessment application site of such drugs.

5. Do the authors consider there is any potential to apply this test to the whole dosage form?

Comments on the Quality of English Language

line 60-61: Change: "is a huge advantage" to "is beneficial"

Reviewer 3 Report

Comments and Suggestions for Authors

Understanding the transportation of NSAIDs through buccal membranes is essential to developing novel formulations. The authors investigated this topic comprehensively in this study. However, they did not describe the relationship between the physicochemical properties of NSAIDs and the results obtained in this study.

The authors should add the consideration and description of the effect of the physicochemical properties such as logP and solubility in water on the transportation through buccal membranes. 

Round 2

Reviewer 1 Report

Comments and Suggestions for Authors

The authors have addressed the previously raised reviewers comments. 

Reviewer 2 Report

Comments and Suggestions for Authors

Thank you for all the provided explanations.

Reviewer 3 Report

Comments and Suggestions for Authors

Since the authors have confirmed that they have made additions and corrections in accordance with the reviewers' suggestions, this paper is acceptable for acceptance.